# Cardiovascular Benefits from Gliflozins: Effects on Endothelial Function

**DOI:** 10.3390/biomedicines9101356

**Published:** 2021-09-29

**Authors:** Teresa Salvatore, Alfredo Caturano, Raffaele Galiero, Anna Di Martino, Gaetana Albanese, Erica Vetrano, Celestino Sardu, Raffaele Marfella, Luca Rinaldi, Ferdinando Carlo Sasso

**Affiliations:** 1Department of Precision Medicine, University of Campania Luigi Vanvitelli, Via De Crecchio 7, I-80138 Naples, Italy; teresa.salvatore@unicampania.it; 2Department of Advanced Medical and Surgical Sciences, University of Campania Luigi Vanvitelli, Piazza Luigi Miraglia 2, I-80138 Naples, Italy; alfredo.caturano@unicampania.it (A.C.); raffaele.galiero@unicampania.it (R.G.); annadimarti@alice.it (A.D.M.); gaetanaalbanese@hotmail.it (G.A.); erica.vetrano@unicampania.it (E.V.); celestino.sardu@unicampania.it (C.S.); raffaele.marfella@unicampania.it (R.M.); luca.rinaldi@unicampania.it (L.R.)

**Keywords:** endothelial dysfunction, gliflozins, cardiovascular diseases, heart prevention

## Abstract

Type 2 diabetes mellitus (T2DM) is a known independent risk factor for atherosclerotic cardiovascular disease (CVD) and solid epidemiological evidence points to heart failure (HF) as one of the most common complications of diabetes. For this reason, it is imperative to consider the prevention of CV outcomes as an effective goal for the management of diabetic patients, as important as lowering blood glucose. Endothelial dysfunction (ED) is an early event of atherosclerosis involving adhesion molecules, chemokines, and leucocytes to enhance low-density lipoprotein oxidation, platelet activation, and vascular smooth muscle cell proliferation and migration. This abnormal vascular phenotype represents an important risk factor for the genesis of any complication of diabetes, contributing to the pathogenesis of not only macrovascular disease but also microvascular damage. Gliflozins are a novel class of anti-hyperglycemic agents used for the treatment of Type 2 diabetes mellitus (T2DM) that selectively inhibit the sodium glucose transporter 2 (SGLT2) in the kidneys and have provoked large interest in scientific community due to their cardiovascular beneficial effects, whose underlying pathophysiology is still not fully understood. This review aimed to analyze the cardiovascular protective mechanisms of SGLT2 inhibition in patients T2DM and their impact on endothelial function.

## 1. Introduction

Type 2 diabetes mellitus (T2DM) is a well-known independent risk factor for atherosclerotic cardiovascular disease (CVD), including coronary, cerebral, and peripheral vasculopathy, a clinical condition that globally represents the worldwide primary cause of complications and death in diabetic patients [1,2,3,4,5,6,7]. Moreover, robust epidemiological evidence indicates heart failure (HF) among the most common CVDs of diabetes, since 1974 when the Framingham study reported a risk of HF in T2DM greater than that of coronary heart disease (CHD) [8,9]. Based on these data, it is imperative to consider the prevention of CV outcomes as an effective goal for the management of diabetic patients, as important as lowering blood glucose.

After worries were raised over the association between myocardial infarction and the PPAR-γ agonist rosiglitazone, the U.S. FDA started to recommend pre- and post-approval studies to demonstrate CVD safety for antidiabetic medications [10,11]. This guidance for the pharmaceutical industry has resulted in the publication, in recent years, of multiple large cardiovascular outcomes trials (CVOTs) that have greatly influenced the diabetes management landscape.

Among the novel glucose-lowering agents approved for clinical use, sodium-glucose cotransporter-2 inhibitors (SGLT2-Is) and incretin-based therapy have been shown to have remarkable CVD beneficial effects. The DURATION-8 study comparing monotherapy versus combined treatment with the SGLT2-I dapagliflozin plus the incretin mimetic GLP-1 agonist exenatide, reported that the co-initiation of these two medicaments improved various glycemic measures and CV risk factors in T2DM patients inadequately controlled by metformin monotherapy [12]. Instead, a superiority of SGLT2-Is with respect to dipeptidyl peptidase-4 inhibitors has been documented in terms of an amelioration of cardiometabolic risk factors and especially of cardiovascular outcomes [13,14].

The high-capacity low affinity SGLT2 receptors, located in the S1 and S2 segments of the proximal convoluted tubules of kidneys, are responsible for >90% of glucose reabsorption from primary urine by transporting glucose across the plasma membrane via a symport mechanism that concomitantly reabsorbs sodium [15,16].

Gliflozins are a novel class of anti-hyperglycemic agents used for the treatment of T2DM that selectively inhibit the SGLT2 in the kidneys, leading to an insulin-independent lowering of blood glucose levels through an enhanced daily urinary loss of up to 100 g of glucose (200–300 kcal) [17]. At the same time, they produce a natriuretic effect by inhibiting sodium reabsorption [18,19].

The design of gliflozins resulted from structural modification of phlorizin, a naturally occurring O-glucoside and non-selective SGLT1/SGLT2 inhibitor. In 2013, the FDA approved the first member of the gliflozin class, canagliflozin, for the treatment of T2DM. Other compounds currently approved include dapagliflozin, empagliflozin, and ipragliflozin. They are well-tolerated drugs that reduce glycosylated hemoglobin (HbA1c) by ~0.5%–0.8% with a low risk of hypoglycemia, decrease BP pressure levels, and induce a weight loss of approximately 2 kg. As predicted by the considerably lower expression of SGLT2 in tissues other than the kidney, the side effects of gliflozins are generally confined to urinary tract infections, although ketoacidosis has been reported and, in 2017, the FDA raised concerns for canagliflozin use over an association of leg and foot amputations [20]. 

Three large CVOTs have demonstrated undeniable CV beneficial effects [21,22,23], respectively for empagliflozin, canagliflozin, and dapagliflozin, that go beyond those expected from simple glycemic control, the optimization of has demonstrated effective CV protection [24,25]. The favorable impact on other conventional risk factors may be involved, such as blood pressure (BP) (3 to 6 mmHg reduction in systolic and 0 to 2 mmHg drop in diastolic BP as consequence of diuretic/natriuretic effect), renal dysfunction (reduced albuminuria by decreased intraglomerular pressure and glomerular hyperfiltration), and body weight (average loss of 1–3 kg) [26,27]. Of importance, the SGLT2-I effect on this last parameter involves visceral adiposity and is not merely a reflection of fluid loss but rather associated with a significant reduction of adipose tissue mass, as demonstrated by bioimpedance spectroscopy [28]. Changes in lipid profile with a decrease in fasting and postprandial triglyceride levels have been described after six months of empagliflozin treatment in T2DM patients with established CHD, whereas a reduction in cholesterol levels has been demonstrated only in some animal studies [29,30,31,32]. The decrease in extracellular fluid and plasma volume with synergistic reduction of both afterload and preload may contribute to cardiovascular benefits, especially in diabetic individuals with impaired function of the left ventricle (LV), CHD, or congestive HF [33]. The importance of the ability of gliflozins to modify various cardiovascular risk factors is indirectly confirmed by the brilliant results of a recent multicenter study, in which a multifactorial intervention was shown to reduce mortality and non-fatal major adverse cardiovascular events (MACE) in diabetic subjects with high cardiovascular risk [34]. Nevertheless, post-hoc analyses of trials have demonstrated that even when adjusting for BP, lipid status, and HbA1c over time, the reduction in CVD death and HF hospitalization is preserved, suggesting the involvement of other mechanisms. Accordingly, a recent analysis of EMPA-REG OUTCOME trial found that the cardioprotective effect of empagliflozin was independent of the achieved level of risk factor control, even if the risk for cardiovascular events was inversely correlated with the number of controlled risk factors at baseline [35]. On the other hand, the modest improvements of cardiovascular risk factors associated to gliflozin use is very unlike to have contributed significantly to the beneficial outcomes in a period of months or few years.

The pathophysiology underlying gliflozin CV actions remains not completely understood, despite the active ongoing research. A lot of contributing mechanisms, both systemic but also direct on vasculature and heart, have been speculated or are still under current investigation. Among these, gliflozins have been associated with the improvement of endothelial dysfunction (ED), a critical initiator of the macro- and micro-vascular complications in T2DM patients with high metabolic risk [36] (Figure 1). 

Here, we provide a summary of the current scientific knowledge on mechanisms of cardiovascular benefits by SGLT-Is, with particular attention to their effects on endothelial function as reported in preclinical and clinical studies.

## 2. Pathogenesis of Endothelial Dysfunction in Diabetes

The endothelium is a thin monolayer covering the inner surface of blood vessels. It represents a barrier between circulating blood and all tissues, responsible for crucial physiological processes that maintain vascular homeostasis [37]. Endothelium secretes a multitude of modulators of vascular tone among which the most important is nitric oxide (NO), synthesized from the amino acid L-arginine by an enzyme expressed in endothelial cells (ECs), the calcium-calmodulin-dependent nitric oxide synthase (eNOS) [38]. In particular, NO exerts pivotal pleiotropic actions against vascular disease, such as relaxation of vascular tone, prevention of monocyte adhesion and platelet aggregation to the endothelium, and inhibition of over-proliferation of vascular smooth muscle cells [39]. Of additional importance, ECs express angiotensin-converting enzyme that play a required step in the conversion of angiotensin I (Ang I) into the biologically active Ang II [40]. 

ED is an abnormal vascular phenotype characterized by a disequilibrium in the synthesis and/or discharge of various endothelial signaling molecules that generates blunted NO formation, oxidative stress, endothelial senescence, and increased microparticles (MPs) shedding. As a result, endothelium-dependent vasodilation is impaired, vasoconstrictor tone enhanced, and leukocyte adhesion and migration increased [41,42]. These features are reflective of the earliest stage in course of atherosclerosis development, initially detectable in highly localized arterial sites exposed to disturbed flow and low shear stress such as bifurcations and curvatures [43].

The literature extensively supports ED as an important risk factor for the genesis of any complication of diabetes, which contributes both to the pathogenesis of macrovascular disease and to microvascular damage [44].

Hyperglycemia is a well-established cause of EC damage and strong evidence suggests oxidative stress, a condition of enhanced formation of reactive oxygen species (ROS) and reactive nitrogen species, and/or of decreased antioxidant potentials, as the cornerstone of ED in the pathophysiology of diabetic complications [45]. Actually, the main mechanism of high glucose (HG)-induced ED is the increased production of ROS. These products may derive from several sources such as an excess of mitochondrial glucose auto-oxidation, functional disturbance of eNOS resulting in the production of superoxide anion rather than NO (so-called eNOS uncoupling), activation of the polyol pathway, generation of advanced glycation end-products (AGEs), and activation of protein kinase C (PKC) [46]. Oxidative stress per se leads to the activation of a series of signaling pathways involved in ED, but also in inflammation, angiogenesis, cellular proliferation, extracellular matrix expansion, apoptosis, and vascular remodeling [47]. In particular, increased ROS production enhances the nuclear translocation of nuclear factor-kappa B (NFkB) and promotes the transcription of pro-inflammatory genes and pro-inflammatory cytokines [48]. Insulin resistance, oxidized-low density lipoprotein, inhibition of AMP-protein kinase (AMPK), and adiponectin may contribute to inflammation during the atherosclerosis development [49].

Briefly, enhanced oxidative stress triggers inflammation, which, in turn, enhances ROS production, with arise of a variety of vicious cycles which intertwine each other and feature a very complex picture. For instance, increased ROS generation alters proteins, DNA, RNA, and lipids, which can cause mitochondrial damage, leading to apoptosis [50]. AGE-induced glycosylation enhances oxidative stress and pro-inflammatory signaling by reducing eNOS activity and positively regulating the expression of inflammatory and prothrombotic genes [51,52]. The activation of NFkB and c-Jun N-terminal kinase (JNK) pathways by oxidative stress contributes to the upregulation of inducible NOS expression, which in turn is involved in insulin resistance and inflammatory processes [53].

Other key features of diabetes further contribute to ED. The impaired insulin signaling reduces eNOS expression, NO production, and the activation of enzymes that regulate eNOS activity, such as phosphatidylinositol-4, 5-bisphosphate 3-kinase (PI3K) and protein kinase B (Akt) [54,55]. High levels of circulating free fatty acids associate with increased lipid peroxidation and NO consumption and interfere with insulin signaling by impairing PI3K activity and reducing NO release, further contributing to ED [56,57,58]. 

In addition to reducing NO bioavailability, diabetes also associates with a decrease in endothelium-derived hyperpolarizing factor (EDHF)-type relaxation and increased endothelial synthesis of vasoconstrictor substances contributing to ED, such as endothelin-1, cyclooxygenase-derived prostanoids, and uridine adenosine tetraphosphate [59,60,61,62]. 

Other factors implicated in the pathogenesis of ED in diabetes are the activation of renin-angiotensin-aldosterone system by metabolic abnormalities (e.g., hyperglycemia, excess free fatty acids, inflammation, and insulin resistance), the endothelial apoptosis and senescence, the dysregulation of microRNAs, and the imbalance of gut microbiota [46].

## 3. Effects of Gliflozin on Vascular Endothelial Function

Impaired endothelial function plays a pivotal role in the pathophysiology of atherogenesis and is usually accompanied by increased oxidative stress and inflammatory responses [63]. In addition to atherosclerosis, ED has been associated with other types of CVD, including HF [64,65,66,67].

Accumulating experimental evidence in animal and in vitro models and related metanalyses [68,69] extensively document that gliflozin may modulate vascular EC activation and endothelial function, even if the expression of SGLT2 receptor on ECs and its role in the control of endothelial function remain interlocutory. Using an organ culture approach with murine aorta rings, immunohistochemistry and transport results showed that ECs per se are able to express the SGLT2 receptor [70]. A recent study further indicated that SGLT1 and SGLT2 protein levels are upregulated ex vivo in pathological arteries of adult rats (i.e., aortic arch exposed to disturbed flow and low shear stress and thoracic aorta exposed to Ang II or endothelial nitric oxide synthase (eNOS) inhibition), being responsible for ED most likely in consequence of the impaired endothelial formation of NO [71]. On the contrary, the inhibition of ROS generation and recovering of NO bioavailability induced by empagliflozin and dapagliflozin, as observed in human coronary arterial ECs (HCAECs) by Uthman et al., could not be attributed to the inhibition of SGLT2 as quantitative PCR to measure mRNA levels of this receptor resulted negative [72]. This observation is congruent with that of Mancini et al., who did not find SGLT2 mRNA in human umbilical vein endothelial cells (HUVECs) and HAECs [73]. 

Acute dapagliflozin treatment induced a direct vasorelaxation of abdominal aortic rings in ex vivo experiments from C57BI/6J mice maintained on a normal chow diet [74], a response not affected by the elimination of endothelium. Thus, an action on vascular smooth muscle cells cannot be excluded, as suggested by the vasodilator effect of dapagliflozin on rabbit thoracic aorta via the activation of protein kinase G (PKG) and thereby of voltage-dependent K^+^ (Kv) channels) [75]. In another study, empagliflozin was found to induce a vasodilation in thoracic aortic rings of rabbits similarly mediated by the activation of PKG and Kv channels, but not related to endothelial cells, other K^+^ channels, the cAMP/PKA pathway, Ca^2+^ channels, or sarco/endoplasmic reticulum Ca^2+^-ATPase (SERCA) pumps [76].

The vasodilator effect of gliflozin could be vascular bed-dependent, as described for canagliflozin and the nonspecific SGLT-I phlorizin, which relaxed pulmonary arteries in a dose-dependent manner but had little or no effect on coronary arteries of C57BI/6J mice [77].

### 3.1. Gliflozins Modulate Endothelial Function by Attenuating Oxidative Stress and Inflammation

The documented ability of SGLT2-Is to prevent adverse vascular alterations, through a direct inhibition of the expression/release of pro-inflammatory mediators or an indirect one via oxidative balance, is very intriguing [78].

#### 3.1.1. Correction of Glucotoxicity

A plethora of evidence suggests that the improvement of ED by SGLT2-Is is mediated by the correction of glucose levels and downstream glucotoxicity involving AGE formation, AGE/RAGE signaling, and then oxidative stress and inflammation, two processes highly involved in ED [79].

An in vitro study demonstrated a protection by SGLT2-Is against high glucose (HG)-induced oxidative stress and mitochondrial dysfunction. In particular, empagliflozin recovered nitrite levels in cultured HUVECs during hyperglycemia, and canagliflozin, dapagliflozin, and empagliflozin, as well as antioxidant gene induction with sulforaphane, prevented HG-induced ED in mouse aortic tissue maintained for 24 h in hyperglycemic medium [70]. 

Many in vivo investigations have used STZ-induced diabetic mice, a well-characterized model of T1DM. In these animals, treatment with empagliflozin reduced blood glucose levels and normalized endothelial function by reversing glucotoxicity (decreased serum level of methylglyoxal and attenuation of AGE/RAGE signaling), oxidative stress (inhibition of NADPH-oxidase), and pro-inflammatory features [80]. In the same model, glycemic control obtained with ipragliflozin administration improved ED, likely by reducing ROS generation as determined by the urinary excretion of 8-oxo-20-deoxyguanosine, and by restoring the phosphorylation of Akt and eNOS^Ser1177^. Concurrently, the expression of inflammatory molecules such as monocyte chemoattractant protein-1 (MCP-1), intercellular adhesion molecule-1 (ICAM-1), and vascular cell adhesion molecule-1 (VCAM-1) decreased [81]. In the same study, in vitro experiments demonstrated the attenuation of Akt and eNOS phosphorylation in HUVECs treated with methylglyoxal, the precursor of AGEs, so the benefit could, at least partially, be attributed to the improvement in HG-associated eNOS dysfunction [81]. 

Other experiments used animal models of T2DM. Empagliflozin administered for six weeks to Zucker diabetic fatty rats (ZDF) reduced glucotoxicity and thereby prevented ED development, attenuated oxidative stress, and exerted anti-inflammatory effects, despite persisting hyperlipidemia and hyperinsulinemia [82]. In another study, T2DM mice treated with dapagliflozin for eight weeks displayed vascular improvements such as lower arterial stiffness and attenuated disfunction of both endothelium and smooth muscle, accompanied by lower circulating levels of glucose and inflammation markers [83].

As a confirmation of vascular benefit through glucose lowering, dapagliflozin or ipragliflozin administered to ApoE^−/−^ mice, STZ-induced diabetic ApoE^−/−^ mice, and diabetic db/db mice via drinking water for four weeks, attenuated macrophage driven atherosclerosis only in diabetic models [84].

Other data suggest that SGLT2-Is may enhance endothelium-dependent vasodilation independently of correction of glycemia, as observed in non-diabetic animals. An acute intravenous administration of canagliflozin in non-diabetic male rats after the onset of coronary ischemia protected against myocardial ischemia-reperfusion injury, enhancing endothelium-dependent vasorelaxation, an effect that could not be mediated by the drop in glucose levels [85]. A study in a spontaneous atherosclerosis model of non-diabetic ApoE^−/−^ mice fed a Western diet indicated that an eight-week treatment with empagliflozin or glimepiride equally decreased blood glucose levels, but atherosclerotic plaque areas in the aortic arch/valve were significantly smaller in the empagliflozin group than in the glimepiride or control groups [86]. These observations strongly suggested the atheroprotective effects of empagliflozin, independent of glucose lowering.

#### 3.1.2. Anti-Inflammatory Effects of Gliflozins

Direct anti-inflammatory actions of SGLT2-Is have been demonstrated in HUVECs using tumor necrosis factor α (TNFα) or interleukin-1β (IL-1β) as inflammatory stimuli. A first study reported that low doses of dapagliflozin exerted direct anti-atherogenic actions in cultured HUVECs. The effect was achieved by the attenuation of TNFα-induced increase in the expression of ICAM-1 and VCAM-1, plasminogen activator inhibitor type 1, and NFκB [74]. Similarly, Mancini et al. documented that canagliflozin inhibited IL-1β-stimulated secretion of MCP-1 and interleukin-6 (IL-6) and reduced their mRNA expression [73]. 

In STZ-induced diabetic ApoE^−/−^ mice fed high-fat diet, a model of accelerated atherosclerosis, dapagliflozin administrated for 12 weeks, partially reversed the formation of atherosclerotic lesions in aorta, as a possible result of the anti-inflammatory effect of the reduction of NLRP3 protein, a component of the NLRP inflammasome, and the consequent reduced secretion of IL-1β by macrophages [87]. Likewise, in a study on nicotinamide/streptozocin (NA/STZ)-treated ApoE KO mice, a seven-day luseogliflozin therapy normalized aortic mRNA of inflammatory genes such as F4/80, TNF-α, IL-1β, IL-6, ICAM-1, PECAM-1, MMP2, and MMP9. In contrast, lipid metabolism-related genes were generally unaffected [88]. In the same study, a six-month treatment significantly attenuated the progression of atherosclerosis in comparison with the severe and widely distributed atherosclerotic changes observed in aortas of luseogliflozin-untreated animals, again without affecting serum lipid parameters. The authors speculated that the rapid appearance of an anti-atherosclerotic effect could be related to the prevention of inflammation rather than the correction of CV risk factors like hyperlipidemia [88]. More recently, in a similar model of STZ-induced diabetic ApoE^−/−^ mice, treatment with empagliflozin significantly suppressed atherogenesis and vascular inflammation as determined by lipid deposition, macrophage accumulation, and inflammatory molecule expression in plaques [31]. In addition, the study demonstrated that empagliflozin reduced the vasoconstrictive eicosanoids derived from arachidonic acid through COX such as PGE2 and TXB2, which are notoriously increased in the diabetic condition [89].

Empagliflozin has been shown to reduce the expression of inflammatory molecules including adhesion molecules and macrophage markers in perivascular adipose tissue, a highly active endocrine organ releasing bioactive substances involved in the development of vascular disease in diabetes [90,91]. Additionally, it decreased the NADPH-oxidase subunits in both aorta and perivascular adipose tissue [31].

In an investigation of non-diabetic ApoE^−/−^ mice by Han et al., empagliflozin was able to mitigate atherosclerosis via improvements in inflammation by systemic or direct effects on vascular cells. In details, circulating levels of TNF-α, IL-6, MCP-1, serum amyloid A, and urinary microalbumin decreased following treatment, and all effects significantly correlated with smaller atherosclerotic plaque areas in the aortic arch/valve. Compared to the control group, empagliflozin also reduced the mRNA expression of TNF-α, IL-6, MCP-1, and the infiltration of inflammatory cells in plaque and adipose tissue, and increased adiponectin circulating levels [86].

A very recent investigation documented that the addition of canagliflozin to the storage solution elicited a downregulation of proinflammatory genes and their products in aortic rings from nondiabetic rats following an in vitro ischemia/reperfusion (I/R) injury [92].

#### 3.1.3. Reduction of ROS Levels by Gliflozins

The mechanism by which SGLT2-Is could directly affect endothelial cell function under inflammatory conditions is not completely understood. An increased AMPK activity through the inhibition of mitochondrial complex 1 has been involved for canagliflozin, although this pathway does not seem influenced by comparable concentrations of dapagliflozin or empagliflozin [73,93]. Some data from animal and cell models suggest that empagliflozin may increase NO production by restoring eNOS activity through enhanced phosphorylation at the activator site Ser1177 of enzyme, but not its transcription rate [81,94]. A similarly increased phosphorylation of eNOS has been described even in diabetic mice treated with ipragliflozin [95].

More convincingly, the correction of ED by gliflozin may by mediated by a reduction in ROS levels, known to be an important hallmark of ED [96]. In a study, HCAECs and HUVECs were pre-incubated with empagliflozin or dapagliflozin and subsequently exposed to TNFα. Both gliflozins completely inhibited the TNFα-induced upregulation of intracellular ROS levels and restored NO bioavailability without affecting eNOS expression or signaling, barrier function, and adhesion molecule expression [72]. In another very recent investigation, HCAECs were pre-incubated with empagliflozin, dapagliflozin, and canagliflozin for two hours, then exposed to a 10% stretch for 24 h. ROS production was increased by stretch but significantly decreased by gliflozins. The inhibition of the sodium-hydrogen exchanger 1 (NHE1) and NADPH oxidases (NOXs) attenuated the stretch-induced ROS production but did not produce a further ROS reduction when combined with empagliflozin. These results seem to indicate that the protection of human endothelial cells by gliflozins is most likely mediated through an anti-oxidative effect that only partially depends on the inhibition of NHE1 and NOXs [97].

#### 3.1.4. Effects of Gliflozins on Glycocalyx Health and Endothelial Senescence

The endothelial glycocalyx is a gel-like layer of glycoproteins, proteoglycans, and associated glycosaminoglycans that covers the vessel internal surface protecting it from inflammatory circulating cells. This dynamic structure constantly sheds and regrows, releases bioactive substances, and triggers biomechanical responses to maintain EC health, particularly mediating the shear-induced release of NO [98]. It has been shown that the enzymatic degradation of glycocalyx components is accompanied by impaired responses to wall shear stress, including reduced NO production [99] through a proinflammatory phenotype [99,100]. A loss of glycocalyx integrity and health may occur after extended exposure to a milieu rich in glucose and ROS. [101,102].

Through a comprehensive set of in vitro experiments on human abdominal aorta ECs subjected to a steady wall shear stress, Cooper et al. demonstrated that empagliflozin exerted in part its atheroprotective effect by improving glycocalyx health through the preservation of glycocalyx integrity, restoration of mechano-transduction response of ECs with damaged glycocalyx, and promotion of an anti-inflammatory endothelium [103]. In a study by Ikonominis et al., T2DM patients treated with GLP-1 agonists, SGLT2-Is, and their combination, achieved after 12 months a greater improvement of endothelial glycocalyx compared to patients treated with insulin. This observation was assessed by PBR (perfused boundary region) of the sublingual arterial micro-vessels, a non-invasive parameter inversely proportional to glycocalyx thickness [104].

A premature senescence of endothelium could contribute to diabetic vasculopathy. Studies in Zucker diabetic rats and in HUVECs cultured on glycated extracellular matrices demonstrated the expression of senescence-associated markers in endothelial cells [105]. EC senescence may be inhibited by NO and likely represents an upstream signaling event to promote ED as indicated by the functional relationship between P53, a factor implicated in impairing endothelial function, and Kruppel-Like factor, a regulator of gene expression essential to the protection of ECs from oxidative stress-mediated injury and apoptosis [106,107]. A recent study found that empagliflozin could attenuate a series of HG-induced effects such as an increase in ECs senescence markers and the downregulation of eNOS expression and NO formation [108]. This result was simultaneously replicated by Park et al. in lean rats where the increased expression of senescence markers in aortic inner curvature, a site at high risk as exposed to a low level of shear stress, was reduced by dapagliflozin to comparable values as those observed in the outer curvature [71].

#### 3.1.5. Gliflozin Effects on Angiotensin System

The angiotensin system is a major contributor to the development of atherosclerotic plaques and to ED [109,110]. Specifically, AngII is known to promote NOX-mediated formation of ROS and vascular pro-oxidant, pro-atherothrombotic, and pro-senescence responses, and to contribute to MPs shedding, effects that ultimately result in ED [111,112,113].

The exposition of porcine coronary artery cultured ECs to HG resulted in an increased protein expression level of ACE and AT1 receptors, both responses abolished by empagliflozin [108]. Since oxidative stress may upregulate the expression of both ACE and AT1 receptors in ECs and vascular smooth muscle cells, a likely interpretation for this effect by empagliflozin was the reduction of glucotoxicity [114]. A recent study by Park et al. extended these findings, observing that AngII was a potent inducer of SGLT1 and SGLT2 in ECs isolated from porcine coronary arteries, independently of hyperglycemia, and that these receptors had a key role to promote ultimately pro-oxidant response and endothelial deleterious effects. Interestingly, both sotagliflozin and empagliflozin prevented these effects [115].

It has been shown that atrial natriuretic peptide (ANP) inhibited the SGLT-2 activity in the kidney [116]. Therefore, it can be speculated that if the transporter would be inhibited via SGLT2-Is, ANP could exert and enhance other functions besides the natriuretic and diuretic ones, like the inhibition of the renin–angiotensin–aldosterone system or protection against AngII-induced cardiac remodeling.

### 3.2. Results from Clinical Studies

ED is among the first disorders that can be detected in atherogenesis by flow-mediated dilation (FMD) of the brachial artery, one of the most widely used tests of endothelial function in macrocirculation and a useful surrogate endpoint, particularly for short-term pharmacological studies [117]. A concomitant measurement of endothelium-derived biomarkers may improve the evaluation accuracy of endothelial function. In this respect, a favorable effect on highly sensitive C-reactive protein, a biomarker of cardiovascular inflammation, has been observed in a phase 3 trial in T2DM patients treated with dapagliflozin, whereas conflicting results have been obtained in the few clinical studies that have investigated FMD in patients treated with this gliflozin [118].

A number of clinical studies exploring the impact of gliflozins on endothelial function have produced results less unequivocal and certainly less strong compared to preclinical investigations.

In a pilot study on 16 T2DM individuals, a two-day treatment with dapagliflozin versus hydrochlorothiazide reduced oxidative stress (defined as a decline in urinary isoprostanes) and produced a significant improvement in measures of vascular and endothelial function, such as differences in aortic pulse wave velocity (PWV), brachial FMD and shear rates, even after correction for mean BP [119]. In a subsequent study, the same authors reported a contradictory null effect on FMD after four weeks of dapagliflozin treatment [120].

In the open-label, single-center DEFENCE study, patients with early-stage T2DM without a known history of CVD, were randomized to receive either 1500 mg/day metformin or 750 mg/day metformin supplemented with 5 mg/day dapagliflozin for 16 weeks. Although FMD tended to improve only in the dapagliflozin group, ΔFMD was comparable between the two drug regimens and only improved in a subgroup of dapagliflozin add-on therapy patients with inadequately controlled disease at baseline (HbA1c > 7.0%). Conversely, urine 8-hydroxy-2′-deoxyguanosin, a biomarker of oxidative stress, was significantly lower in the dapagliflozin group compared to the metformin group [121].

A small group of 54 patients with uncontrolled T2DM began a therapy with add-on dapagliflozin (5 mg/day) or non-SGLT2 inhibitor drugs to improve HbA1c. After six months, the glycemic control and the microvascular endothelial function assessed by reactive hyperemia peripheral arterial tonometry (RH-PAT) were significantly improved in the dapagliflozin group. In addition, dapagliflozin significantly reduced the abdominal fat mass [122].

In the EDIFIED study involving high-risk diabetic patients due to underlying CHD, a 12-week therapy with dapagliflozin in addition to insulin and metformin failed to demonstrate a significant improvement of FMD, despite a worsening trend of this parameter within the placebo group [123]. A later study of T2DM patients with no history of CVD demonstrated that a six-month dapagliflozin therapy resulted in a significant improvement of central PWV, probably caused by the reduction of body fat, especially visceral fat, rather than an amelioration of vascular function [124,125].

The recently published ADDENDA-BHS2 aimed to evaluate whether dapagliflozin, regardless of its glucose-lowering effect, could mitigate the ED and increase the endothelial resilience to I/R injury in T2DM patients with atherosclerotic disease [126]. The study demonstrated that a 12-week course of treatment improved vasomotor function in both macro- and microcirculation as compared to glibenclamide, in a setting of comparable glucose lowering between the two groups. Consistently, NO production during FMD as indicated by plasma nitrite levels, increased after treatment with dapagliflozin but not with glibenclamide. Instead, the change in FMD after I/R was not different between the two regimens. In an intragroup exploratory analysis, the study demonstrated that dapagliflozin improved FMD in individuals with severe arterial wall dysfunction when compared to glibenclamide, as well as when compared with baseline values [126]. The positive results on FMD obtained in this study likely depend on the recruitment of patients with more severe arterial wall dysfunction than prior trials and on a carefully balanced reduction in blood glucose in the two study arms. The improved vascular function was congruent with results reported by Solini et al. in the resistive index measured in the renal artery of patients under dapagliflozin treatment [119].

Other clinical studies have employed empagliflozin. Sawada et al. studied 50 patients with established CHD treated with empagliflozin 10 mg/day. Six months of therapy resulted in a statistically significant reduced C-reactive protein level and a significantly improved FMD. The recovery of endothelial function was strongly associated with a reduction in plasma triglycerides [29].

Two studies explored the potential effect of treatment with empagliflozin on arterial function in T1DM. The first reported a decline in arterial stiffness in a group of young patients [127]. The second showed that empagliflozin, metformin, and empagliflozin/metformin significantly improved FMD to a comparable level, but that arterial stiffness maximally decreased in the group treated with the empagliflozin on top of metformin [128], a drug that has also been shown to exert important extra-glycemic cardiovascular protective effects [46,129,130].

An exploratory study demonstrated a beneficial in vivo effect of empagliflozin on shear stress and endothelial function in T2DM subjects [131].

Of note, the EMBLEM trial designed to elucidate potential mechanisms by which empagliflozin protects CV systems, failed to demonstrate in 105 T2DM patients with established CVD an improvement in peripheral endothelial function measured by RH-PAT after 24 weeks of empagliflozin treatment [132]. This negative result was confirmed by a secondary and exploratory analysis of the trial [133].

The ongoing PROCEED trial aims to examine the effects of ipragliflozin on ED in T2DM patients suffering from chronic kidney disease [134].

## 4. Glifozins and Heart Failure

Based on gliflozin positive findings from three main CVOTs, as well as from trials investigating CV benefits in chronic HF and two large multinational studies, SGLT2-Is have been identified as a new class of compounds for the treatment of diabetic patients suffering from HF, a common and serious comorbidity of T2DM whose prevention represent a crucial therapeutic goal [21,22,23,135,136,137,138,139,140,141].

Several possible explanations are suggested for gliflozin benefits in HF. 

Merely, they correct factors predisposing or aggravating HF. Their pharmacological effects including osmotic diuresis induced by glycosuria and concomitant increased Na^+^ excretion, decreased BP and, more importantly, improved fluid overload with greater loss in interstitial relative to intravascular volume can reduce preload and LV filling pressures in HF patients [142,143]. Moreover, gliflozins may modulate the afterload through the reduction of arterial BP and stiffness [127,144,145].

The inhibition of SGLT2 may attenuate the downstream effects of neurohormonal activation in HF and ameliorate the tissue oxygenation thanks to an increase in hemoglobin and hematocrit levels [146,147].

Above all, gliflozins are being shown to exert numerous favorable actions on the heart, such as the amelioration of LV mass and hemodynamics, reduction of myocardial inflammation, oxidative stress and fibrosis, and direct pleiotropic effects on cardiomyocytes, although there is an apparent absence of detectable SGLT2 expression in the heart [16,148,149].

Based on recent evidence, a key point in the pathophysiology of EHpHF is the ED in microvascular coronary bed, another mechanism which may be beneficially affected by gliflozins (Figure 2).

### 4.1. Gliflozins Correct Endothelial Dysfunction in Microvascular Coronary Bed

The impairment of endothelial integrity or function may represent a pivotal feature in HF, as indicated by high circulating levels of apoptotic endothelial-derived MPs, a surrogate biomarker and effector of ED, in non-diabetic patients with chronic HF [150]. Indeed, a clinical study showed that both ED and abnormal vascular structure, respectively indicated by FMD and intima-media thickness, were present in HFpEF and could contribute to its pathogenesis and maintenance [151].

Due to strict anatomical connection, a sick endothelium can affect cardiomyocyte health. Thus, the amelioration of endothelial function in coronary microcirculation may represent a key mechanism among those suggested in the protection of failing heart by gliflozins.

The mechanism by which SGLT2-Is could affect the function of cardiac microvascular endothelial cell is still debated [152]. It has been reported as an effect mediated by the inhibition of NHE1 in cardiomyocytes [153,154].

A study of the hearts of STZ diabetic mice revealed that empagliflozin improved myocardial structure and function by preserving cardiac microvascular barrier, sustaining eNOS phosphorylation and endothelium-dependent relaxation, as well as improving microvessel density and perfusion. These effects likely resulted from the inhibition of mitochondrial fission mediated by an AMPK-dependent mechanism inhibiting Drp1 activation [155].

In a rodent model of early diabetes, a 10-week treatment with empagliflozin improved coronary flow velocity reserve and fractional area change (a systolic contractility index) monitored with sequential measurement by non-invasive Doppler ultrasound imaging. The concomitant increase in l-arginine/ADMA ratio as an indicator of increased NO production, possibly indicated an improvement of cardiac contractile function via an NO-dependent improvement of endothelial coronary microvascular function [156].

Interesting results were obtained by Juni et al. using a novel co-culture system of human cardiac microvascular endothelial cells with adult rat ventricular cardiomyocytes. The study provided the first evidence that cardiac microvascular ECs modulated cardiomyocyte contraction and relaxation, an effect that was lost after pre-incubation with TNF-α but restored by empagliflozin through the reduction of ROS mitochondrial production and cytoplasmic accumulation, and by recovering endothelial NO bioavailability [81]. In a study on a similar model, the same investigators showed that uremic serum from patients on dialysis impaired endothelium-mediated enhancement of cardiomyocyte relaxation and contraction, an effect that was rescued by empagliflozin [157]. These results may provide a novel mechanism linking cardiac microvascular ED to the pathogenesis of HF in chronic kidney disease.

### 4.2. Gliflozin Benefit in HFpEF

The paradigm of HFpEF, a condition that accounts for a significant proportion of HF in the Western world, has shifted over the past years from a mere cardiomyocyte disease to a disorder that initially involves cardiac microvascular ECs. Currently, HFpEF is described as a systemic syndrome mediated by risk factors and co-morbidities, often of metabolic nature, resulting in a multi-organ pro-inflammatory state and leading to myocardial dysfunction and profibrotic remodeling through a cascade of events that initially involves cardiac microcirculation [158]. In this context, coronary microvascular dysfunction represents a prevalent phenotype and a powerful predictor of HFpEF, and ED a key determinant of its outcome [159,160,161].

An optimal level of intracellular NO might be critical for the proper function of cardiomyocytes, and nitrosative stress has been implicated as a master mediator in the pathophysiology of HFpEF [162]. Based on these premises and experimental findings, the improvement of myocardial microvascular EC function and amelioration of oxidative stress may represent the pivotal mechanisms by which SGLT2-Is prevent HF.

In a non-diabetic rat model of hypertension and diastolic dysfunction, which develops a series of metabolic, cardiac, and renal disturbances commonly observed in HFpEF patients, dapagliflozin mitigated some features of endothelial activation and dysfunction at the coronary level such as the upregulation of VCAM-1 and E-selectin, the downregulation of eNOS, and an increased expression of NFκB [163]. 

Two small investigations support these results in humans. In 184 diabetic subjects with HFpEF randomly divided into three groups (empagliflozin, luseogliflozin, and tofogliflozin), FMD was significantly higher and E/e’ (echocardiographic index of LV diastolic dysfunction) significantly lower at the end of a 12-week period of treatment, with a significant association between ∆mean E/e’ and ∆FMD [161]. In a pilot study evaluating 26 patients with T2DM and heart disease, a six-month treatment with tofogliflozin significantly ameliorated LV end-diastolic dimension but this change was not associated with a modification of %FMD, perhaps because this parameter was not suitable for the evaluation of small vessels, such as coronary microcirculation [164]. Incidentally, FMD was significantly improved only in patients with average HbA1c levels >8.0%, a result similar to the DEFENCE study reporting restoration of ED by dapagliflozin especially in diabetic patients with high HbA1c levels [121].

## 5. Conclusions

The inhibitors of SGLT2 are the currently used drugs most recently added to the therapeutic armamentarium of T2DM, which have proven to have intriguing CV beneficial effects beyond the anti-hyperglycemic ones.

Various large CVOTs have demonstrated the non-inferiority or, more importantly, the CV superiority of gliflozins in people with diabetes. Based on these results, the management landscape for these patients has gained an actual potentiality to improve both the quality and expectancy of life for these patients.

SGLT2 inhibition represent a very attractive therapeutic option even on the grounds of its demonstrated benefits in non-diabetic HF patients, especially those with HFpEF, a challenging syndrome that until now lacked effective treatments.

The current experimental evidence mainly displays the anti-oxidant and anti-inflammatory actions of gliflozins, both at the macrovascular and microvascular level, including the coronary microvascular endothelium. The modulation of these and other pathways ultimately improves endothelial function, the first step involved in the development of atherosclerosis and the impairment of heart function, two key determinants of future CV events. As supported by several preclinical investigations but only by some clinical trials, correction of ED is an important mechanism among those involved in favorable CV outcomes brought on by gliflozin.

Further studies need to evaluate both safety and clinical efficacy to better understand the impact of gliflozins on frail elderly subjects characterized by numerous comorbidities [165].

We hope that future research can resolve knowledge gaps and further clarify the pathophysiological mechanisms and target sites of action, in order to determine how these compounds maintain vasculature and heart health, and how they can be implemented into personalized therapy programs to allow for the greatest benefit for patients.

## Figures and Tables

**Figure 1 biomedicines-09-01356-f001:**
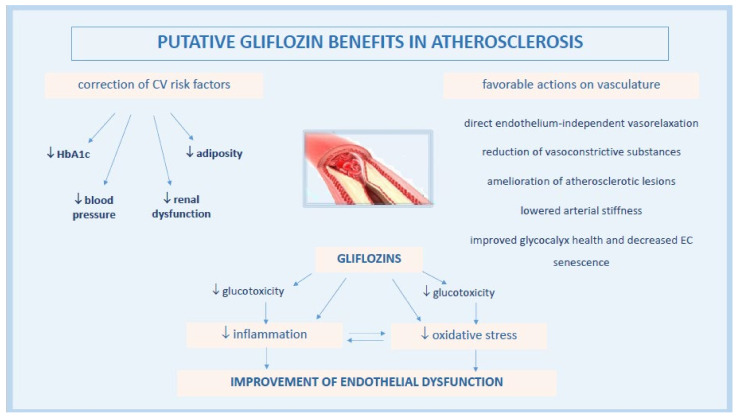
Pathophysiological mechanisms of anti-atherosclerotic actions by gliflozins.

**Figure 2 biomedicines-09-01356-f002:**
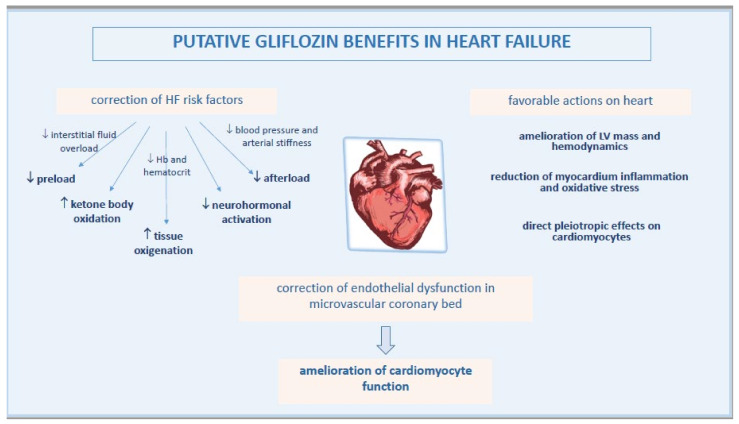
Pathophysiological mechanisms of improved heart function by gliflozins.

## Data Availability

Not available.

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
