# Peer review of "Cardiovascular Benefits from Gliflozins: Effects on Endothelial Function"

_biomedicines, 2021, doi:10.3390/biomedicines9101356_

Round 1
Reviewer 1 Report
Thank you for the opportunity to review the paper "Cardiovascular benefits from glyflozins: effects on the endothelial function".
The paper needs extensive language revision to improve readability.
I also feel that the approach of the revision does not respect the title and subject matter of the SI to which it is submitted
I would eliminate chapter 2 (LARGE CARDIOVASCULAR OUTCOME TRIALS WITH GLIFLOZINS), chapter 3 (CORRECTION OF CARDIOVASCULAR RISK FACTORS BY GLIFLOZINS) and chapter 6 (GLIFLOZINS AND HEART FAILURE) and expand chapters 4 and 5.
These suggestions serve to differentiate this work from the many reviews published to date on this topic.
Author Response
Reviewer 1
Thank you for the opportunity to review the paper "Cardiovascular benefits from gliflozins: effects on the endothelial function".
Authors: We would like to thank the reviewer for his/her helpful comments which enabled us to improve the manuscript.
The paper needs extensive language revision to improve readability.
Authors: The paper was reviewed by a native English speaker.
I also feel that the approach of the revision does not respect the title and subject matter of the SI to which it is submitted
I would eliminate chapter 2 (LARGE CARDIOVASCULAR OUTCOME TRIALS WITH GLIFLOZINS), chapter 3 (CORRECTION OF CARDIOVASCULAR RISK FACTORS BY GLIFLOZINS) and chapter 6 (GLIFLOZINS AND HEART FAILURE) and expand chapters 4 and 5.
These suggestions serve to differentiate this work from the many reviews published to date on this topic.
Authors: We have eliminated chapter 2 (LARGE CARDIOVASCULAR OUTCOME TRIAL WITH GLIFLOZINS), whereas chapter 3 (CORRECTION OF CARDIOVASCULAR RISK FACTORS BY GLIFLOZINS) has been inserted in a shortened form in the current chapter 1 (INTRODUCTION).
We felt we had to maintain the chapter 6 (GLIFLOZIN AND HEART FAILURE, current chapter 4) since the demonstrated involvement of gliflozins have been involved in the correction of endothelial dysfunction in microvascular coronary bed, a factor which plays a key role in the pathophysiology of heart failure.
Moreover, we largely rewritten the remaining sections to expand the topic of gliflozin effects on endothelial dysfunction. Thus, the paper structure resulted extensively modified, and 24 additional references were included.
Reviewer 2 Report
This is a well done and comprehensive review paper. It reads well. It is well constructed. My only critique is the length of presentation. You review all the various mechanisms of ED development which are well known. I would cut this down considerably and concentrate on what FLOZINS particularly do to ameliorate these factors. So you can say that reduction of ROS leads to less ASHD. Flozins reduce ROS by x, y, z. Likewise data such as the effects of flozins on cardiomyocytes is of interest and should be given more space in the paper. In short, make the paper focused.
I have no COI
No plagiarism
no self citations
No ethical issues
Author Response
Reviewer 2
This is a well done and comprehensive review paper. It reads well. It is well constructed. My only critique is the length of presentation. You review all the various mechanisms of ED development which are well known. I would cut this down considerably and concentrate on what FLOZINS particularly do to ameliorate these factors. So you can say that reduction of ROS leads to less ASHD. Flozins reduce ROS by x, y, z. Likewise data such as the effects of flozins on cardiomyocytes is of interest and should be given more space in the paper. In short, make the paper focused.
Authors: We would like to thank the reviewer for his/her helpful comments which enabled us to improve the manuscript.
Following your advice, we have erased chapter 2 (LARGE CARDIOVASCULAR OUTCOME TRIAL WITH GLIFLOZINS) and inserted the chapter 3 (CORRECTION OF CARDIOVASCULAR RISK FACTORS BY GLIFLOZINS) in a shortened form in the current chapter 1 (INTRODUCTION). Moreover, we largely rewritten the remaining sections to expand the topic of gliflozin effects on endothelial dysfunction. The article structure resulted extensively modified, and 24 additional references highlighted in yellow were included.
Reviewer 3 Report
In this manuscript, the authors summarized the benefits of Gliflozins on Type 2 diabetes mellitus associated cardiovascular disease. Gliflozins are selectively inhibit the sodium glucose transporter 2 in the kidneys to lower blood glucose, which further improve endothelial function. The authors discussed the benefits of endothelial cells for T2DM after treatment with Gliflozins, then discussed potential benefits for cardiovascular system. Diabetic cardiomyopathy is characterized by cardiac hypertrophy, cardiac lipidtoxicity, fibrosis, and diastolic dysfunction with normally preserved ejection fraction. The authors discussed potential cardioprotective mechanisms of Gliflozins for diabetic cardiomyopathy. The authors should increase more content for discussion of benefits of cardiovascular system by Gliflozins. It should be noted that most benefits of Gliflozins may be indirect on cardiomyocytes, since whether SGLT2 is expressed in cardiomyocytes remain unclear.
Author Response
Reviewer 3
In this manuscript, the authors summarized the benefits of Gliflozins on Type 2 diabetes mellitus associated cardiovascular disease. Gliflozins are selectively inhibit the sodium glucose transporter 2 in the kidneys to lower blood glucose, which further improve endothelial function. The authors discussed the benefits of endothelial cells for T2DM after treatment with Gliflozins, then discussed potential benefits for cardiovascular system. Diabetic cardiomyopathy is characterized by cardiac hypertrophy, cardiac lipidtoxicity, fibrosis, and diastolic dysfunction with normally preserved ejection fraction. The authors discussed potential cardioprotective mechanisms of Gliflozins for diabetic cardiomyopathy.
The authors should increase more content for discussion of benefits of cardiovascular system by Gliflozins. It should be noted that most benefits of Gliflozins may be indirect on cardiomyocytes, since whether SGLT2 is expressed in cardiomyocytes remain unclear.
Authors: We are very grateful for your positive judgement, and we accepted willingly your suggestion to better focus the paper content. Therefore, we erased chapter 2 (LARGE CARDIOVASCULAR OUTCOME TRIAL WITH GLIFLOZINS) and inserted the chapter 3 (CORRECTION OF CARDIOVASCULAR RISK FACTORS BY GLIFLOZINS) in a shortened form into the current chapter 1 (INTRODUCTION). Moreover, we largely rewritten the remaining sections to expand the topic of gliflozin effects on endothelial dysfunction. The article structure resulted extensively modified, and 24 additional references highlighted in yellow were included.
Reviewer 4 Report
Comments to Authors:
The manuscript need extensive re-writing to publication standard.The authors may work on revision extensively.
Major Comments:
Abstract: The abstract is repharse or re write in a scientific manner. For example see the sentence "Endothelial dysfunction (ED) is an abnormal vascular phenotype characterized by a disequilibrium in the synthesis and/or discharge of various endothelial signaling molecules" It is poor definition scientific writing!!!!
It should write as" Endothelial dysfunction (ED) is an early event of atherosclerosis which involving adhesion molecules,chemokines and leukocyte to enhance low-density lipoprotein oxidation,platelet activation,ascular smooth muscle cell proliferation and migration"
2. You should include gliflozins structure, mechanism of action, enlisted published work related to this drugs?
3. Please discuss any clincal trials associated with this drugs?
Author Response
Reviewer 4
The manuscript needs extensive re-writing to publication standard. The authors may work on revision extensively.
Authors: We would like to thank the reviewer for his/her helpful comments which enabled us to improve the manuscript.
Following your suggestion, the paper has been extensively rewritten. In particular, we erased the chapter 2 (LARGE CARDIOVASCULAR OUTCOME TRIAL WITH GLIFLOZINS) and inserted the chapter 3 (CORRECTION OF CARDIOVASCULAR RISK FACTORS BY GLIFLOZINS) in a shortened form into the current chapter 1 (INTRODUCTION). Moreover, we largely rewritten the remaining sections to expand the topic of gliflozin effects on endothelial dysfunction. The article structure resulted extensively modified, and 24 additional references highlighted in yellow have been inserted.
Major Comments:
Abstract: The abstract is repharse or re write in a scientific manner. For example see the sentence "Endothelial dysfunction (ED) is an abnormal vascular phenotype characterized by a disequilibrium in the synthesis and/or discharge of various endothelial signaling molecules" It is poor definition scientific writing!!!!
It should write as" Endothelial dysfunction (ED) is an early event of atherosclerosis which involving adhesion molecules, chemokines and leukocyte to enhance low-density lipoprotein oxidation, platelet activation, vascular smooth muscle cell proliferation and migration"
Authors: The abstract has been rewritten following your advice.
- You should include gliflozins structure, mechanism of action, enlisted published work related to this drugs?
- Please discuss any clincal trials associated with this drugs?
Authors: Responses to point 2 and 3. We considered that the discussion of the structure and mechanism of action of gliflozins, as well as the discussion of any clinical trials associated with these drugs, was not inherent in the topic of paper, as indicated also by other reviewers who suggested to delete the chapter on trials, and focus this review exclusively on endothelial dysfunction.
Round 2
Reviewer 1 Report
The authors adressed the majority of the reviewer comments and the paper is improved
In reviewer opinion chapter four must be modified according to the title of the special issue and should be describe only the effecst of SGLT2-i on endothelial dysfunction
Author Response
We wish to thank the reviewer for the comment. We have modified the chapter 4, according to the title of the special issue. Accordingly, we changed references in main text in order.
Reviewer 4 Report
The revised manuscript may be acceptable
Author Response
The revised manuscript may be acceptable
Reply: We wish to thank the reviewer for the comment.
